# Moral Distress and Moral Agency: Staff Experience of Supporting Self-Determination for People with Dementia

Cecilia Ingard [1,*], Maria Sjölund [2] and Sven Trygged [1,*]

1  Department of Social Work, Criminology and Public Health, Faculty of Health and Occupational Studies, University of Gävle, SE-80176 Gävle, Sweden
2  Department of Social Work, Umeå University, SE-90187 Umeå, Sweden; maria.sjolund@umu.se
*  Correspondence: cecilia.ingard@hig.se (C.I.); sven.trygged@hig.se (S.T.)

**Abstract:** People living in nursing homes have the right to self-determination, and difficulty in accommodating this right can create moral distress in staff. This study aimed to explore experiences of situations of moral distress and to identify nursing home staffs' needs to act with moral agency. Six group interviews were conducted with nursing home staff. Content analysis of the interview responses showed that moral distress can be rooted in both concrete situations with residents and factors related to the work environment and policy requirements. Personnel can address moral distress through both active and passive means. Staff acted to address moral distress mainly in situations with residents and sometimes in relation to co-workers, but they did not try to influence the policy level.

**Keywords:** moral distress; nursing homes; dementia; staff experiences; resident autonomy; self-determination; working conditions; moral agency

## 1. Introduction

Stress in working life can be caused and experienced in many different ways. This study will focus on a special kind of stress, namely, moral distress, experienced when staff strive to promote self-determination among nursing home residents living with dementia. Moral distress is psychological distress experienced in human service organizations and can appear when staff feel they cannot do their work as well as they want (Lützén and Kvist 2012). In a systematic review, Morley et al. (2019) concluded that moral distress comprises both psychological distress and experiences of events with a moral valence.

The Swedish national guidelines for elderly care (National Board of Health and Welfare (NBHW 2012)) underscore the ethical foundation guiding residential care in Sweden. Staff members are responsible for considering residents' ability to influence their care and for adapting support to meet everyone's needs in nursing homes. The Swedish welfare system focuses on the person with dementia's *right to* self-determination rather than *how* the person should be supported (Nedlund and Taghizadeh Larsson 2016). Caregivers are obliged to protect persons with dementia from harm and simultaneously respect their free will (Giertz et al. 2019). However, it is hard for persons with dementia to be autonomous and make decisions. Therefore, formal and informal caregivers are crucial in supporting the person with dementia's opportunities for self-determination and, at the same time, protecting that person from harming him/herself (Haugen et al. 2019).

The fundamental principle of self-determination for nursing home residents relies on staff having sufficient time to listen to and foster trusting relationships with residents. However, there is a need for greater emphasis on how organizations support staff so they can enhance residents' self-determination (Lindmark et al. 2022; Szebehely 2016). Residential care organizations must recognize the importance of staff well-being alongside resident-centered care (Szebehely 2016). Moreover, despite efforts to promote resident participation, nursing homes have struggled to achieve high levels of involvement among

individuals with dementia (Ingard et al. 2023). When staff members cannot deliver care in line with policy guidelines and their own perception of quality care, they may experience moral distress.

Understaffing and limited resources such as time, activities, and materials contribute to this distress, and family members sometimes fail to provide necessary supplies when residents move into nursing homes (Pijl-Zieber et al. 2018). However, moral distress can also catalyze staff members to act as moral agents to advocate for residents' rights (Banks and Gallagher 2009; Jameton 2017; Lützén and Kvist 2012). Ethical issues related to communication with residents, which is crucial for protecting their autonomy but requires additional time, and the risk of burnout among staff members caring for individuals with dementia have also been identified (Preshaw et al. 2015). Without organizational support, staff have limited opportunities to promote resident self-determination (Woods 2019).

Nevertheless, the extent to which residents can influence the care they receive depends on several factors. Organizational factors such as leadership, care culture, staff ratio, and involvement of relatives play a role in determining staff ability to facilitate resident autonomy and self-determination. Adequate skills are crucial for staff to involve residents in decision-making (Strøm and Slettebø 2021). Broad self-determination for persons with dementia in nursing homes is uncommon; it requires that the staff and managers prioritize efforts to promote it (Ingard et al. 2023). Staff perceptions of self-determination for the residents are intertwined with their perceptions of what constitutes reasonable care. As well, how individual staff members understand resident participation differs (Erlandsson et al. 2022).

## 1.1. Previous Studies of Moral Distress and Agency

Moral theories strive to systematically describe the nature of morality and many such theories stipulate how humans should live morally correctly (Rachels and Rachels 2010). Rachels and Rachels (2010) frame the ethics of care as a virtue ethics (the development of virtuous character traits). There are other ethical currents, among them utilitarianism (maximizing overall happiness and minimizing harm), and idealistic ethics (which emphasizes adherence to moral rules and duties). We assume that idealist ethics is the most relevant in this context (The Internet Encyclopedia of Philosophy 2024). The individuals' ideals and values connect to how they act in particular situations and handle moral uncertainty and social pressure. Thus, addressing moral distress requires a multi-faceted approach that combines individual reflection, organizational support, and a commitment to ethical practices—recognizing the social dynamics contributing to moral distress at work (The Internet Encyclopedia of Philosophy 2024; Mander 2016).

Lützén and Kvist (2012) differentiated among moral distress (i.e., psychological distress resulting from being unable to act according to one's internal moral guidelines), moral stress (i.e., resulting from moral demands and lack of control), and stress of conscience (i.e., a philosophical perspective involving moral trouble). These perspectives are linked to the ethical climate in an organization and to the organization's moral sensitivity to staff. According to Banks and Gallagher (2009), strict policies, understaffing, administrative work, communication challenges, and insufficient training can contribute to moral distress in care. Emotions such as disgust, shame, or guilt feelings can affect staff, in line with their moral compass and self-evaluation of how things should be. Morley et al. (2019, p. 655) delineate various definitions of moral distress, with a common foundational tenet asserting that moral distress arises when "a moral judgment has been made and there are institutional constraints that prevent that moral judgment from being acted on." However, delineating moral judgment poses challenges, as terms such as belief and awareness can influence interpretations of moral distress. Moral beliefs, awareness, and conflicts may be synonymous with moral distress. Furthermore, narrative frameworks of moral distress often encompass psychological ramifications.

Staff experience of being unable to provide proper care due to institutional rules can cause moral distress and suffering, such as self-blame, self-doubt, and, in the end, burnout.

However, staff play a crucial role in creating the institutional culture (Banks and Gallagher 2009). Mänttäri-van der Kuip (2020) conceptualized moral distress within social work, emphasizing its relevance to understanding moral suffering among staff. According to Jameton (1984, p. 6), "moral distress arises when one knows the right thing to do, but institutional constraints make it nearly impossible to pursue the right course of action." Furthermore, moral distress must be set in a broader context, taking account of factors such as the institutional context, colleagues, and the public health and welfare system, all of which affect experiences of moral distress. Staff often have alternative choices and are responsible for them (Jameton 1984).

Jameton (2017) also acknowledged that moral distress sometimes can have positive aspects when staff members are striving to address moral dilemmas. Moral agency refers to a person's capacity to act in a morally relevant manner and can apply to staff moral engagement in their work (Lützén and Kvist 2012). The individual is a moral agent if he/she can make moral judgments about right or wrong actions and has moral motives (Beauchamp and Childress 2009). Moreover, Banks and Gallagher (2009) discussed staff as moral agents who perceive situations and feel, think, and act according to ideals and emotions. However, sometimes they do not have appropriate feelings toward care recipients (Banks and Gallagher 2009). Staff, as moral agents, are not the only actors responsible for ethical dilemmas, and it is also important to pay attention to the political and organizational context. Staff must know about the global, national, and local policy documents shaping their practice. Professionals such as social service staff and nurses see themselves as moral agents with power and often make changes at the individual level rather than the organizational or political level (Banks and Gallagher 2009). Nevertheless, efforts to address moral dilemmas need to be implemented through collaboration among different care professionals and involve ongoing discussions of what it means to be a good staff member working in care (Banks and Gallagher 2009).

In sum, moral distress refers to the psychological and emotional discomfort experienced by individuals when they cannot act in alignment with their moral or ethical values due to external constraints or conflicting situations. This phenomenon is often encountered in various professional settings, such as healthcare and social services. In this study, the most relevant aspect of moral stress relates to interpersonal relationships in the workplace (where individuals may feel pressure to compromise their ethical principles for the sake of organizational goals or expectations) or when organizational policies or procedures conflict with an individual's ethical beliefs.

Furthermore, for this study, we chose to interpret the informant's narrative according to the Morley et al. (2019, p. 1) definition, which stipulated that moral distress could be a "combination of (1) the experience of a moral event, (2) the experience of 'psychological distress"; moreover, (3) a direct causal relation between (1) and (2) together are necessary and sufficient conditions for moral distress". When we write about a moral event we use the words situations, circumstances and interactions.

### 1.2. Moral Distress in Nursing Homes

According to VonDras et al. (2009), nursing home staff sometimes view residents as challenging to deal with, mainly due to problematic behaviors such as aggressiveness. Negative attitudes toward residents were associated with lower job satisfaction among staff. Stressors experienced by nursing home staff include concerns about respecting residents' autonomy, acting reasonably, and fulfilling professional duties. Additionally, co-workers can experience stress in these environments (VonDras et al. 2009). Furthermore, demands from relatives for prolonged life care instead of focusing on quality of life can contribute to moral distress (Midtbust et al. 2022). Nursing home staff often feel compelled to provide care they consider inappropriate and incompatible with their understanding of reasonable care. The COVID-19 pandemic has heightened staff experiences of moral distress, with factors such as lack of activities and restrictions on visiting amplifying ethical

dilemmas, as staff are required to follow policy guidelines that may potentially harm residents (Haslam-Larmer et al. 2023).

Edberg et al. (2008) highlighted the need for ethical discussions among nursing home staff. Due to time limitations, staff members face challenges in adhering to ethical care guidelines and may struggle to find solutions that provide optimal care. When residents do not consent, staff actions may be seen as coercive and restraining. Most studies of moral distress focus on care rather than explicitly addressing nursing home staff perspectives (Spenceley et al. 2017). Nursing home staff sometimes follow their moral compass, only to clash with regulations, causing them to experience moral distress (Kälvemark et al. 2004). Furthermore, Kälvemark et al. (2004) shed light on the need for better organizational support in addressing moral dilemmas. Professional duties such as feeling obliged to comply with the employer's requirements to take shortcuts affect the care and the staff experiences of moral distress (VonDras et al. 2009).

In summary, there are many studies regarding moral distress in care, but not so many in nursing homes. There is a need for research on how the demands placed on staff to promote resident autonomy can contribute to moral distress and to staff acting as moral agents when they cannot fully adhere to policy guidelines or institutional rules and approaches regarding resident participation.

### 1.3. Aim

This study aimed to explore experiences of situations of moral distress and to identify nursing home staffs' needs to act with moral agency.

## 2. Methods

### 2.1. Settings

Over the past few decades, the organization of residential care in Sweden has undergone significant changes, with the prevailing norm being that older individuals should reside in their own homes while receiving support from the public welfare system and other available resources, such as relatives or privately purchased services (Ullmanen and Szebehely 2014). Entry into a nursing home typically requires severe disability or dementia on the part of the care recipients. Moreover, nursing home residents need support from staff and relatives to handle their daily lives (SOU 2017, p. 21). In nursing homes, residents can expect to have apartments where they can decide on the furnishings and how to live their daily lives (NBHW 2016). If needed, a person whom the resident chooses can make decisions instead of the person him/herself (NBHW 2012). According to policy documents, staff should strive to promote residents' self-determination and autonomy (Lindmark et al. 2022). The norm in Swedish residential care, as described by Elmersjö (2020), is helping residents to maintain their capabilities, and staff should take responsibility for the residents' best interests, although residents sometimes disagree with that approach. Accordingly, the relationship between residents and staff is crucial to foster autonomy in persons with dementia (Haugen et al. 2019).

### 2.2. Participants

The empirical material for this study consists of six group interviews with 24 nursing home personnel, including assistant nurses and occupational therapists, conducted in 2021 and 2022. We excluded temporary staff, who mostly work on an hourly basis, from the interviews. The informants' experience of working in the profession ranged from one to 32 years. We recruited participants from three different municipalities in Sweden and included both municipally operated nursing homes and those operated by for-profit organizations.

### 2.3. Data-Collection

We first contacted nursing home managers and then presented the study to the staff during an information meeting attended by those available in the nursing home at that moment. Staff registered their interest in the interviews by telling the researcher (or

managers, if they could not decide right away) after the meetings. When staff members could not attend the information meeting, the managers sometimes asked them if they were interested in participating in the interviews. The interviews, conducted by the first author, took place in a separate room in the nursing homes. All interviews were in Swedish; later, the second author translated the relevant quotations into English. We asked participants about their work with residents' self-determination and autonomy. The interviews lasted 45 min to two hours and were recorded and transcribed. We obtained signed consent forms from the participants before the interviews.

*2.4. Analysis*

The perspective of the analysis is to understand (a) circumstances, situations and interactions that may lead to moral distress and (b) how moral distress can affect the moral agency of staff when they strive to promote residents' self-determination.

The interviews were qualitatively analyzed thematically, following the approach outlined by Graneheim et al. (2017). Our interview guide included questions about how staff experience working to promote residents' autonomy and self-determination. Please see Table 1:

**Table 1.** Location, participants, and length of group interviews.

| Owner of Nursing Home | Informants | Length of Interview |
|---|---|---|
| No. 1. Municipal (Smalltown) | 3 women, aged 49–63 years, assistant nurses | 2 h |
| No. 2. For-profit (Midtown) | 2 women, aged 39 and 53 years, assistant nurses | 45 min |
| No. 3. Municipal (Midtown) | 6 women, aged 37–49 years, assistant nurses | 1 h and 10 min |
| No. 4. For-profit (Midtown) | 5 women, aged 32–59 years, assistant nurses | 1 h |
| No. 5. Municipal (Bigtown) | 2 women and 1 man, aged 24–55 years, assistant nurses | 50 min |
| No. 6. For-profit (Bigtown) | 4 women and 1 man, aged 31–59 years, occupational therapist and 4 assistant nurses | 1 h and 45 min |

We interpreted the interviews and used content analysis focusing on understanding and describing moral distress and moral agency. In the analysis, both subject and context were essential to the purpose of this study. We conducted the analysis inspired by Graneheim and Lundman (2004), and Graneheim et al. (2017), we selected text that we interpreted as related to moral distress and agency. Then, we divided the text into meaning units and condensed them. We named them with codes. Moreover, we interpreted the codes in relation to moral distress and moral agency. Based on the codes, we formulated six categories, and from those categories, we built three main categories: (1) Interactions contributing to staff acting against their beliefs, (2) working environment contributing to staff acting against their beliefs, and (3) force staff to act against the regulations to promote self-determination.

The categories represented different situations, interactions, and circumstances that we interpreted as related to moral distress. When analyzing the interviews, the first author read the transcripts several times and divided the text into units of meaning, i.e., segments of critical situations, circumstances, and interactions that can cause moral distress and result in personnel sometimes having to exercise moral agency. In addition, the first author labeled the text segments with codes and sorted the codes, categorizing them based on similarities and interpreting them according to their relevance to moral distress. The other authors also read the transcripts; all authors discussed the codes and categories and jointly decided which categories were relevant to the study. The authors divided the categories into three descriptive main categories. The main categories and categories can be described as having a low level of abstraction (close to the informant descriptions) but a high degree of interpretation when we interpret the story according to moral distress (Graneheim et al. 2017).

*2.5. Ethics*

We do not name the municipalities, nursing homes, or participants in this study in order to anonymize the results. To identify the responses, we use the names Smalltown, Midtown, and Bigtown and the interview number. The staff members participated voluntarily in the group interviews, and informed consent was obtained from all subjects involved in the study. The study was conducted in accordance with the Declaration of Helsinki and was approved by the Swedish Ethical Review Authority (Dnr 2021-00067).

## 3. Results

Moral distress in nursing home care contained experiences of lack of time, high external demands and non-care related tasks hindering staff to give good care and handling aggressive residents and family members. This triggered staff to act as moral agents. The number after each quotation indicates the specific group interview. Please see Table 2:

**Table 2.** Meanings units, condensed meanings units, interpretation/categories, and main categories.

| Meaning Units | Condensed Meaning Unit Description Close to the Text | Condensed Meaning Units Interpretation of the Underlying Meaning/Categories ("What") | Main Categories ("How") |
|---|---|---|---|
| you have 12 [residents] and one sits and shilly-shallies for a long time, so it takes time because it's not like they do it any faster. And you know, you have three [residents] screaming outside … yes, it is very stressful. | you have three [residents] screaming outside … yes, it is very stressful. | Routines clash with residents' self-determination. | Interactions contributing to staff acting against their beliefs. |
| When the man passed away … they notified the nurse and those who were working that he was palliative … that is, dying, but she wanted to send him to the hospital. And he died somewhere in the corridors instead of dying at home in bed … she was hysterical … and then relatives like them have a lot of power, things can go so wrong. | she was hysterical … and then relatives like them have a lot of power, things can go so wrong. | Relatives' demands affected the care. | Interactions contributing to staff acting against their beliefs. |
| … we have a lot of tasks that don't really concern the care and nursing profession. So I think that if more assistant nurses were allowed to focus on working as assistant nurses, it would be a somewhat attractive workplace … and you wouldn't wear yourself out so quickly. | we have a lot of tasks that don't really concern the care and nursing profession. | Counteract the work for self-determination. | Working environment contributing to staff acting against their beliefs. |
| One would wish that they [i.e., policymakers] could come and look one day and see how it is. Because I don't think they know … But that's not how reality works, but that's the price we have to pay. Like me, as an assistant nurse, because we have to make sure it's nice. | One would wish that they [i.e., policymakers] could come and look one day and see how it is. | Demands in policy-documents. | Working environment contributing to staff acting against their beliefs. |
| It [i.e., self-determination] can sometimes feel like it clashes with certain laws that exist regarding how to treat a dementia patient, and so on … It sometimes feels like those who decide, perhaps even higher up, may not have insight into how it works in reality. | clashes with certain laws that exist regarding how to treat a dementia patient. | Act according to their beliefs. | Force staff to act against the regulations to promote self-determination. Main category 3 is further elaborated in Table 3. |

*3.1. Main Category 1: Interactions Contributing to Staff Acting against Their Beliefs*

3.1.1. Routines Clash with Residents' Self-Determination

Nursing home residents can contribute to moral distress among staff when there are emerging problematic situations and staff cannot work according to what they have learned to do in those situations, or when the residents want to oppose the institutional approach and rules. Do staff act as moral agents and ignore rules that hinder the residents from doing what they want to do? Staff can also respond to situations to protect themselves from burnout. Striving to enhance the residents' autonomy and self-determination can be time consuming, as the residents themselves are supposed to manage this, trying to handle everyday tasks to stimulate their capabilities and frame their autonomy. When staff are stressed because of the situation, the residents sense that and may undertake tasks, such as getting dressed and brushing teeth, even more slowly than usual:

> It can take longer when they are supposed to keep doing things on their own . . . you have 12 [residents] and one sits and shilly-shallies for a long time, so it takes time because it's not like they do it any faster. And you know, you have three [residents] screaming outside . . . yes, it is very stressful. (No. 3)

Residents could stress the staff if they did not want to cooperate on a given day and were in a bad mood. Some residents would prefer to live someplace other than the nursing home, which affects their interaction with staff. Sometimes residents did not get along with each other, and the staff needed to mediate between the residents to help resolve their issues. Moreover, residents could be unstable, and in those cases, it was scary for the staff to leave the residents alone. Staff needed to be patient in their interactions with the residents:

> I think it's very stressful if you believe they're going to fall or if two residents who can't stand each other get too close . . . well then you have to step in between . . . you need to solve it quickly . . . and unfortunately, it can happen that they hit each other. (No. 1)

Sometimes there were situations when staff felt unsafe because the residents were acting in a threatening way. However, our informants said that dangerous situations mostly arose when the staff were stressed. In certain situations, the residents all needed help at the same time, and in those cases, staff must prioritize some residents while others need to wait:

> Sometimes if something happens, then two things happen at the same time. Which is not so strange. Here we have a disaster and there we have a disaster. Then it can happen that the person you meet when you run from one to the other doesn't get treated well. You try to look happy, but you may not have time to stay as you would have liked. (No. 2)

The staff could not force residents to do things, and sometimes the residents' choices could appear unethical, but the residents should be able to exercise their free will. According to NBHW (2012), nursing homes in Sweden are obliged to promote the residents' autonomy and opportunities for self-determination regardless of dementia. In that case, staff could experience moral distress, fearing that people outside the nursing homes could blame them. However, sometimes the staff contravened institutional rules in favor of the residents' well-being and, in those cases, were probably acting as moral agents. In the quotation below, the staff member let the resident dress as he/she wanted instead of following the institutional rules:

> But Gosh, they live here! It doesn't really matter if you eat breakfast in your nightgown one day. Or walk around in your pajamas at night. I mean, you do that yourself at home. This is their home. (No. 6)

An example of a situation that caused staff moral distress was when two residents fell in love with each other and wanted to have sex. The staff experienced that situation as problematic because, although the residents were adults and had free will, they sometimes did not understand the situation because of their dementia:

> And then we had this, we had two residents here and they kind of fell in love and they wanted to have sex—and stuff like that. We had a lot of discussion about that . . . no, we can't stop them as long as both of them are into it. (No. 6)

The staff experienced a lack of time to facilitate residents' autonomy, which affected the treatment of the residents and troubled the staff members' consciences. The staff said that working with persons with dementia takes a long time, and it would have gone faster if they performed the chores themselves and did not involve the residents in them:

> Our professional ethics are constantly falling apart. So, you don't actually get to do the job you would like to do, but it turns out to be half bad . . . if you work with dementia, you know that it has to take time. (No. 1)

Activities based on the resident's unique interests were rare because of a lack of time. The resident has, for example, the right to go outside the nursing home, but the staff did not have time to facilitate that, so the outdoor activities could be brief. Some nursing homes promised to allow the residents to go outside every day, but some residents did not want to go outside:

> Again, this matter of adaptation [to institutional conditions] that—hell, you [i.e., the resident] must be satisfied if you get to walk around the home, like that, because that's what I have time for. The person [i.e., the resident] may be used to being out for an hour and a half every day, but there is no such possibility today, we have to go in now and it's terrible. (No. 1)

### 3.1.2. Relatives' Demands Affected the Care

Relatives do not always know what is best for their next of kin, and sometimes the residents are not the same people as they were before dementia. However, the staff said it was crucial to pay attention to the residents' wishes instead of the relatives' points of view. Some relatives also wanted to decide about medication, and the staff did not appreciate that. The staff said they often knew the residents better than the relatives did because they interacted with them daily. There were situations when relatives contributed to moral distress. One example was when a relative forced the staff to send a dying man to the hospital instead of letting him die in his home at the nursing home. The man died alone in the corridor at the hospital:

> When the man passed away . . . they notified the nurse and those who were working that he was palliative . . . that is, dying, but she wanted to send him to the hospital. And he died somewhere in the corridors instead of dying at home in bed . . . she was hysterical . . . and then relatives like them have a lot of power, things can go so wrong. (No. 6)

It is also stressful when relatives do not care about their next of kin. Nevertheless, the staff rationalized that and thought that the relationship between the resident and the relatives might have been poor before the resident was institutionalized, and that the staff should not blame the relatives. Some staff believed that the relatives simply left their next of kin in the nursing home and did not care about them. Relatives could be upset when visiting their relatives with dementia. In some cases, the staff even had to deal with the relatives' emotions:

> It's difficult, just this, with relatives . . . and there will be a lot of guilt and a lot of emotions in relatives . . . and then it will be we, the staff, who have to take it. It doesn't have to be directed at us. (No. 4)

### 3.2. *Main Category 2: Working Environment Contributing to Staff Acting against Their Beliefs*
### 3.2.1. Counteract the Work for Self-Determination

The most common staff experience was that the understaffed workplace created stress in their daily work. The staff had to trust one another. Their colleagues would not think one staff member was lazy because she/he took more time with one resident, because that resident demanded more time:

> We can trust one another. I know that she'll take care of the whole department, if necessary, if I stay [with a resident]. Because she knows that I don't do it because I want to be lazy and stay there, but because this person needs someone with them. (No. 6)

Some colleagues did not work in a way that facilitated the residents' autonomy, in which case some staff would then act as moral agents and correct these staff:

> "I also usually tell the staff at once if they are behaving a little [inappropriately]". (No. 5)

The regular staff do not always appreciate their temporary staff: they do not know the residents, do not have many responsibilities, and may speak Swedish badly. However, they also are conscious that they need temporary staff during vacations.

The working conditions imposed many demands that limited staff abilities to give the residents meaningful daily lives. Staff also experienced that if they could not live up to the high demands of other professional groups, such as managers and nurses, they experienced stress. According to the informants, limited staff and therefore time was the most common factor contributing to poor working conditions:

> "You really want to do a little more, and then it's time with them [i.e., the residents] and more staff" (No. 2)

According to the staff, they had insufficient time to talk with one another, which could lead to care mistakes, and it is essential to have enough staff to care for persons with dementia:

> We can brighten up their daily lives—"It's their sentence, they won't get out of here"—you can say it's their last stop. And to be able to meet all these demands that come from all sides from occupational therapists . . . the boss, the residents . . . the relatives. (No. 1)

Moreover, when staff were absent, the residents could not do activities but would stay in the sitting room doing nothing:

> "You can't do much. Sometimes there are two of us in the afternoon. You can only sit with them". (No. 5)

The staff did not blame the residents for the poor working conditions, which were because they had to perform many tasks that were assigned as assistant nurses' duties:

> It's never them [i.e., the residents] who are a concern or, like, what you need to do with them. No, no, it's really just one's own time . . . we have a lot of tasks that don't really concern the care and nursing profession. So I think that if more assistant nurses were allowed to focus on working as assistant nurses, it would be a somewhat attractive workplace . . . and you wouldn't wear yourself out so quickly. (No. 2)

Stress accumulated when the staff lacked time for the most crucial work and could not give the residents sufficient emotional support or care to brighten their everyday life. Sometimes staff worked overtime to help their colleagues, thereby acting as moral agents. There were situations when the staff were so stressed that they took their work mentally home with them, pondering the situation:

> And it turned out like this because I . . . was about to bang my head against the wall completely. And then there was more of this when I got home. So I thought—if I've done it, I've given medicine, did I do it? And then I called my colleges . . . am I starting to get demented? Yes, God, stress can do a lot to your head. (No. 1)

However, the staff also learned how to set boundaries around their work to protect themselves as agents:

> "You want everyone [i.e., the residents] to do well, and you want to do extra, and you do the extra at your own expense, if you don't learn to set limits". (No. 1)

During the COVID-19 pandemic, staff needed to isolate the residents, so they could not move around the nursing homes as they wanted to. Some staff said that they collaborated with their colleagues. When they got tired, they asked another staff member to take over because they needed help to deal with the residents correctly:

> And if you've kept going, then, like finally, you tell your colleague you're taking over now, because of this frustration, like now I've done everything ... I feel now I have to have a break. (No. 1)

### 3.2.2. Demands in Policy-Documents

According to the informants, the policymakers are too far away from the floor in nursing homes (No. 2). The staff cannot follow the routines and ideas about residents' opportunities for self-determination stipulated in the documents because they are not grounded in reality. In that case, the staff interpreted the policies in a way that worked on the floor, thereby acting as moral agents:

> But guidelines and such things are not really the reality. One would wish that they [i.e., policymakers] could come and look one day and see how it is. Because I don't think they know ... But that's not how reality works, but that's the price we have to pay. Like me, as an assistant nurse, because we have to make sure it's nice. (No. 2)

There is the routine, for example, that all residents should eat in the sitting room. Nevertheless, some staff problematized that and reminded us that the residents rent their rooms in the nursing home and, for that reason, must have opportunities to decide how they will eat their meals:

> They are demented, then you take them out of their own homes and they have to live with complete strangers. And then they have to eat breakfast with their neighbors, and then the staff think that everyone has to come out and eat in the dining room ... no, it's a dilemma. (No. 6)

Moreover, some staff problematized the standardized routines, meaning that all interaction should be unique. With standardized practices, the residents are not individuals. Instead, staff are supposed to follow standard routines, and not care that residents are individuals with unique needs. In some nursing homes, the policy documents were more visible than in others. For example, one organization put up memory patches related to the guidelines for elderly care on the wall, even in the toilet. Moreover, staff have a willingness to make use of and improve the policy documents:

> It's fun to do something else, like [work with the value-based ethics from the guidelines]. But then it's always good, I think, to be a little reminded of that ... yes, but it's there, then you figure it out. (No. 3)

We created Figure 1 based on the results of our interviews. The content analysis showed that there were factors at different levels that could lead to moral distress, and the figure is an attempt to systematize and make these findings visible. The figure shows factors and situations that can contribute to moral distress and moral agency at different levels of society. The inner circle shows interactions with the residents that contribute to staff experience of moral distress. The next circle shows factors in the institution that affect the situation and contribute to moral distress. The next circle shows local policy documents that demand that staff strive to promote the residents' self-determination and autonomy, and the last circle shows the political ambitions regarding self-determination and autonomy stipulated in the law and national documents. All levels affect the staff and contribute to moral distress, and result in staff sometimes acting as moral agents.

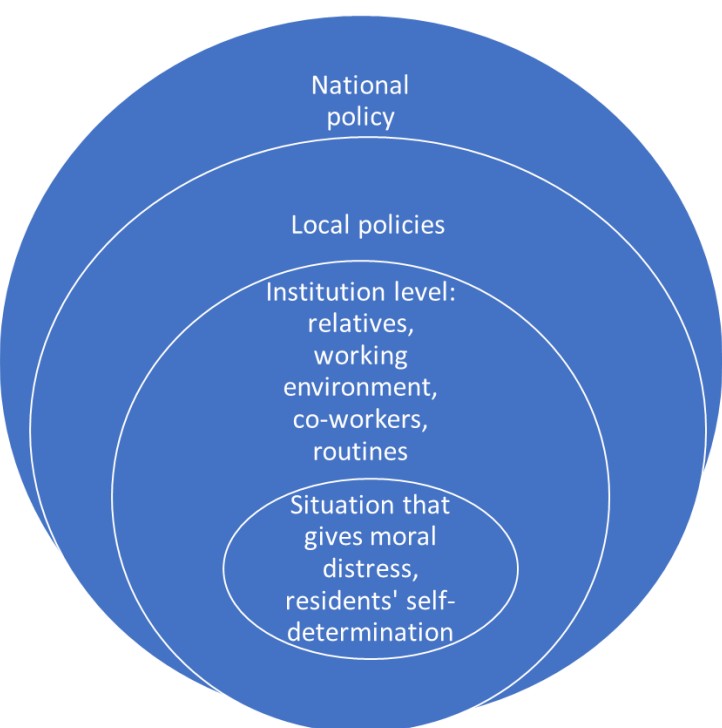

**Figure 1.** Moral distress grounded in different levels.

*3.3. Main Category 3: Force Staff to Act against the Regulations to Promote Self-Determination*
Act According to Their Beliefs

Based on the interviews, we interpreted some situations as the staff exercising moral agency. In Table 3, we show how we interpreted the situation by presenting part of our content analysis.

**Table 3.** Analysis of moral agency in response to moral distress at different levels.

| What They Say | Condensation/Key Words | Passive/Active Dimension | Agency Level |
|---|---|---|---|
| I think it is important that they [i.e., the residents] get to decide something . . . you must be strong when you then report to your colleagues, that now I have done this. I know it's wet in the incontinence pad, but he didn't want me to change it. | did not want me to change/did not want | Passive agent | Situation at individual level causes moral distress; staff member does not act, i.e., a passive agent |
| I usually tell the staff right away if they are behaving a bit [inappropriately]. | tell staff if they behave/tell staff | Active agent | Factor that causes stress at the institutional level; staff member acts as active agent |
| But guidelines and such things are not really the reality. One would wish that they [i.e., policymakers] could come and look one day and see how it is. Because I don't think they know . . . But that's not how reality works, but that's the price we have to pay. Like me, as an assistant nurse, because we have to make sure it's nice. | because we have to make sure it's nice/we have to make | Active agent | Factors causing moral distress at local policy level; staff as active agents doing what works best in the nursing home |
| It [i.e., self-determination] can sometimes feel like it clashes with certain laws that exist regarding how to treat a dementia patient, and so on . . . It sometimes feels like those who decide, perhaps even higher up, may not have insight into how it works in reality. So sometimes you can't one hundred percent stick to exactly all the frames, so you have to think pretty much outside the box. | so you have to think pretty much outside the box/think outside the box | Active agent | Factors at national policy level causing moral distress; staff acting as active agents to make it work in the nursing home |

Here, we present examples of staff whom we interpreted as acting as moral agents at different societal levels (see Figure 1). Some of the examples illustrate how moral agency is sometimes passive, for example, when choosing not to follow an institutional rule because of residents' preferences. We also identify situations in which the staff were more active agents, doing things for the residents' best interests instead of following institutional rules and approaches, policy documents, or laws because the policymakers did not understand daily life in the nursing homes and the rules did not support the residents' daily life. Table 3 presents examples from our content analysis of the interviews regarding moral distress as resulting in—as we determine it—moral agency. The table includes one example from each of the four different societal levels: individual, institutional, local, and national (see Figure 1).

## 4. Discussion

In this study, according to the aim, we identified situations (mostly interaction with stakeholders) and factors (mostly related to working conditions) that caused staff experiences of moral distress, resulting in staff acting with moral agency. Such situations were related to the residents' demands to be autonomous, as required in policy documents, and to the limited resources given to the care sector so it can meet those requirements. Factors contributing to moral distress were relatives, working conditions, untrained colleagues, nursing home routines, and policy documents with high ambitions about residents' rights.

According to Jameton (1984), moral distress can arise when the staff know the right thing to do, but the institutional rules hinder them from doing so. Furthermore, Jameton (1984) shed light on the broader context and identified factors that hinder staff from acting based on moral distress. Most of our informants were conscious of the residents' right to self-determination. They expressed moral distress when they could not act to promote that, for example, by accompanying the residents outside the nursing home. According to Erlandsson et al. (2022), staff believe that resident participation is intertwined with perceived good care. According to Banks and Gallagher (2009), emotions such as shame can be a driving force of staff in their daily work, but if institutional hindrances cause staff to act against their feelings, it contributes to moral distress. Staff who found that the institutional rules did not help residents could have feelings of self-blame and burnout linked with moral distress (Banks and Gallagher 2009). In our study, the situation regarding moral distress among staff was found to be most common in interactions with residents. The staff were supposed to work for the residents' self-determination and autonomy, but they could not because of institutional factors.

Moreover, moral distress can catalyze staff to act to promote morality rights (Lützén and Kvist 2012). According to Jameton (2017), staff who identify moral dilemmas can strive to influence society and policy documents as moral agents. Strict policies, understaffing, administrative work, and insufficient training can hinder staff from working according to their moral compass (Banks and Gallagher 2009), and those factors were also evident in our study. When we analyzed our interviews, we interpreted some situations as the staff acting as moral agents. However, they acted only at the individual level, and had fewer ambitions to act at the organizational or political levels. The staff encountered ethical dilemmas in their daily work on the floor, and for that reason may have found it easier to act as moral agents at the individual and institutional levels. Furthermore, staff may not be invited to reflect on the policy level. The policy levels apply a top–down perspective (as our informants experienced it) and may not invite staff input in the same way as do the individual and institutional levels, which apply more of a bottom–up perspective. According to Banks and Gallagher (2009), it is harder for staff to affect the institutional or political level as moral agents. Staff can, however, discuss ethical dilemmas with other professionals and affect the climate and working conditions in the nursing home at the institutional level (Banks and Gallagher 2009).

Acting as a moral agent could be complex, and there are not always right or wrong answers as to what is the right action. In Table 3, we presented a quotation about a

staff member who did not change incontinence pads because the resident did not want them changed. In this case, the staff member acted as a moral agent but against the institution's rule stipulating that nobody should walk around in wet incontinence pads. Furthermore, we can regard this staff member as a *passive* moral agent who did not act due to the resident's wishes. The staff member did nothing actively to follow the resident's wishes, and because of that, we labeled him/her as a passive agent. If the staff member allows the resident to walk around in wet incontinence pads, the other residents may experience a bad smell. We have identified a knowledge gap concerning moral agency as consisting of subgroups, such as passive and active moral agents. Moreover, the quoted assistant nurse was a passive agent, simply following the resident's wishes and opposing the institutional rules. However, a passive agent can also be framed as a person who prioritizes the resident's wishes.

In Table 3, we presented an example of moral agency that we can interpret as *active* agency. A staff member told another staff member that he/she was not working correctly to facilitate the resident's self-determination. That staff member did something actively as a moral agent. The literature praises moral agency as good for care (Banks and Gallagher 2009; Jameton 2017). However, there are ethical dilemmas for staff when they are required to act as moral agents and uphold residents' wishes according to policy documents but against the institution's rules and approaches. Nevertheless, not following the institutional rules and letting residents' wishes guide them could lead to harmful and unethical situations for the residents.

The policy documents that steer nursing homes express high ambitions about the residents' right to self-determination. However, there should be more discussion in these documents about supporting staff when endeavoring to promote resident autonomy (Lindmark et al. 2022). According to Ingard et al. (2023), nursing homes have not managed to facilitate self-determination for persons with dementia to any great degree, and staff must apply the approach that self-determination is essential. In our study, some of the staff mentioned that the policy documents have unrealistic ambitions given the resources allotted (e.g., staffing limits). Our informants said that the policymakers needed to learn how things work on the nursing home floor, so that the policy documents have legitimacy for staff.

There is a gap between the ideas about residents' self-determination stipulated in policy documents and the reality in nursing homes, and this gap can contribute to moral distress. It is hard to achieve ambitious goals set forth in policy documents regarding resident self-determination and autonomy because staff need more resources to pursue that goal and more time for ongoing discussion of related ethical issues. However, nursing home managers have the opportunity to influence the policy levels and function as a link between the work on the floor and the policymakers. The task of reducing staff moral distress must fall to nursing home managers and organizations (Kälvemark et al. 2004). The demands articulated in policy documents regarding resident self-determination may contribute to staff moral distress. It is good that policy documents should articulate high aspirations. However, the documents could also be a legitimate guide for staff work if nursing home staff were more involved in their development and implementation.

## 5. Limitations

In some cases, we recruited the informants by asking the nursing home managers to suggest appropriate staff as participants (if the staff could not attend the information meeting), which could lead to selection bias. Furthermore, the managers attended the same information meeting as the staff when the staff announced their interest in participating. The study is based on interviews with direct care staff and does not include department heads or any other managers who might have complementary opinions. The group interviews differed in how much the informants talked, with informants in the smaller groups speaking more freely.

## 6. Conclusions

Demands for self-determination by nursing home residents can lead to moral distress for staff, mainly because of unforeseen situations that staff cannot handle because of limited time and staffing. Factors such as relatives' demands, poor working conditions, untrained colleagues, institutional rules and approaches, and strict policies all contribute to moral distress. The gap between the reality on the floor and the ambitions articulated in policy documents could probably be remedied if staff were given more resources.

The empirical data come from Swedish nursing homes. However, despite organizational differences between countries and regions, many of the present findings are likely applicable to other settings.

An example of a situation that leads to the exercise of moral agency by staff is when residents want something that goes against institutional rules. However, if staff follow residents' wishes, then staff must be ready to defend their decision to do so. Acting as a moral agent was most common at an individual level, and staff did not try to affect the organizational or political levels to improve care.

Acting as a moral agent is often good but can contribute to ethical dilemmas. Staff can act as passive moral agents (i.e., not acting at all), thereby not helping to remove residents from an unfavorable position by following their wishes.

**Author Contributions:** Conceptualization, C.I., M.S., and S.T.; methodology, C.I., M.S., and S.T.; software, C.I.; validation, C.I., M.S. and S.T.; formal analysis, C.I., M.S. and S.T.; investigation, C.I.; resources, C.I., M.S. and S.T.; data curation, C.I., M.S. and S.T.; writing—original draft preparation, C.I., M.S., and S.T.; writing—review and editing, C.I., M.S. and S.T.; visualization, C.I., M.S. and S.T.; supervision, M.S. and S.T.; project administration, S.T.; funding acquisition, S.T. All authors have read and agreed to the published version of the manuscript.

**Funding:** This research received no external funding.

**Institutional Review Board Statement:** The study was conducted in accordance with the Declaration of Helsinki, and approved by the Institutional Review Board (or Ethics Committee) of Etikprövningsmyndigheten I Uppsala protocol code Dnr 2021-00067, date of approval 17 February 2021, Etikprövningsmyndighetens Marcus Edelgård.

**Informed Consent Statement:** Informed consent was obtained from all subjects involved in the study. Written informed consent has been obtained from the patient(s) to publish this paper.

**Data Availability Statement:** The data presented in this study are available on request from the corresponding author due to ethical reasons.

**Conflicts of Interest:** The authors declare no conflicts of interest.

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
