# Peer review of "Moral Distress and Moral Agency: Staff Experience of Supporting Self-Determination for People with Dementia"

_socsci, doi:10.3390/socsci13050237_

Round 1
Reviewer 1 Report (Previous Reviewer 3)
Comments and Suggestions for Authors
Paper presented addresses a topic of great interest: moral distress and moral agency. In this sense, it is a novel research that provides valuable information to the scientific community. The article provides a brief explanation of some ethical currents that help to understand the conception of morality used by researchers. Reference is also made, albeit slightly, to the social importance of morality. In this sense, I suggest to the authors that they should not forget that the social perspective is fundamental to clarify moral presuppositions. Regarding moral aspects, the authors use as reference an entry in a virtual encyclopedia. Since the research is primarily psychological, this brief reference may be sufficient. However, it is important that first level sources are used. On the other hand, it is also important that the authors are aware of the need to analyze moral presuppositions. Individuals and social groups, depending on our culture, the society in which we live or our social class, are more likely to approach one moral approach or another. These aspects are explained in the text in an excessively brief way. However, I believe that they may be sufficient to meet the minimum objectives of the research. In any case, I kindly recommend the authors to keep these issues in mind for other research on moral aspects. Despite these questions and suggestions, I believe that these aspects have been minimally clarified in the article. The methodology is clear and well structured. The results are appropriate, address the stated objectives and are understandable. The discussion is correct and the conclusions are correct. Therefore, this is a work of interest that has been substantially improved. Congratulations to the authors.
Author Response
Thank you for your comments. We have added one more reference regarding ethics. Best regards authors.
Reviewer 2 Report (New Reviewer)
Comments and Suggestions for Authors
I believe the manuscript has improved, particularly the introduction, but still needs revision of clarity of the phenomena, structure of the methods and an adequate categorisation in the result section before publication in Social Sciences.
1. INTRODUCTION.
The introduction has indeed improved, but now too long to be clear and readable. Recommendation to shorten at least 1/3 of the text and write more succinct and clear. Think, what is your key messages. For instance, the following sentence is not clear to me and please omit your coming results. “In this study, we will adopt this definition of moral distress. It links to how the staff experienced moral events that led to psychological distress as we interpreted the interviews.” Furthermore, see below regarding the aim. If you want to use moral events, you need to define what they are.
Additionally, you need to write a clear rationale for the study in the end of the introduction. I suggest a paragraph starting with: In summary.... and finish with a rationale (there is a need of...), so the reader clearly understand your aim. How about moving the follwoing section before the aim and reformulate “focus” to need to study? "The core focus is to understand a) circumstances (moral events) that may lead to moral distress and b) how moral distress can affect the moral agency of staff when they strive to promote residents’ self-determination". See also the part after the aim “1.4. Nursing Homes in Sweden” below under METHOD.
2. AIM.
The aim is not clear as the phenomena you study is not clear. The parenthesis confuses and you have as I mentioned above not defined what you mean with "moral event". You need to work on the aim and clarity regarding of the phenomena in your analysis part and in the result section, see below.
3. METHODS.
This part is still unstructured. Suggestion to start with declaring the design of the study and use the following headings to guide me and the readers: Setting, Participants, Data-collection, Analysis. Regarding Setting, I suggest to rename “1.4. Nursing Homes in Sweden” to Setting and only describe short the setting for your study. Other descriptions outside your study, suggestion to either omit or move to introduction.
Regarding Analysis section, you have worked good, but I am still confused regarding your perspective of analysis and the phenomena you study. Your write “describing moral distress and moral agency”, “interpreting moral distress”, ”capture these dilemmas”, “their work with residents' self-determination and autonomy” “critical situations and factors (moral events) can cause moral distress and result in personnel sometimes having to exercise moral agency”. I think this is a good sentence describing your analysis: “labeled the text segments with codes and sorted the codes, categorizing them based on similarities and interpreting them according to their relevance to moral distress”. However, here you state moral distress and is this your phenomena you study, how informants interpret moral distress? Please clarify your perspective and also revise the following phrases to more adequately describe your steps: “interpreting … through latent content and condensed meanings” (it is not clear what you mean), “The authors divided the categories into two descriptive subthemes: moral distress and moral action.” (I don’t understand according to Graneheim and the resultsection do not give clarity. Additionally, recommendation to omit “abductive” as you don’t explain what you mean and not how you have done.
4. RESULTS
You start with “this section describes moral events that can lead to moral distress and trigger the staff to act as moral agents”. Please start with an answer to a revised aim instead and suggestion to list revised subthemes (or themes and subthemes) as your mainresult. You have still not revised your categorisations. In qualitative results the headings should be your themes/subthemes. Please make a table of your categorisation table of your subthemes (see Graneheim 2004 column of subthemes in Figure 3 for guidance. I think this can make you understand that your headings are not subthemes, think that the aim guides the formulation of them. For instance, do you think this reflects the work of categorisation? “Sub-team: moral distress 3.1. Situations that Contribute to Moral Distress (Individual Level: Distress Related to Staff Interactions with Residents)”. I think you can use Figure 1 in your Discussion instead.
Author Response
Thank you for your comments on the article. Please see the uploaded document. / Best regards authors.

Round 2
Reviewer 2 Report (New Reviewer)
Comments and Suggestions for Authors
Dear authors,
Thank you for improved introduction to your manuscript. However, I still don't understand your result. In order to be published I recommend to work on the following:
|
AIM |
I am sorry, but I still can’t find the answer to your new aim in the results. Your subthemes are not answers to your aim, instead repetition of your focus (circumstances, situations and interactions). The reader need to know WHAT are the circumstances, situations and interactions (give meet to the bones, that is, what do the circumstances, situations and interactions contain?). In the discussion you start with:
“In this study, we identified situations and factors that caused staff experiences of moral distress, resulting in staff acting with moral agency.” So isn’t your aim: to explore experiences of situations of moral distress and to identify nursing home staffs’ needs to act with moral agency. Furthermore, the core focus, I think it should be reformulated to perscpective of analysis and to be moved to the analysis section.
|
Methods Please omit or reformulate to describe your analysis of different levels: “However, moral distress and moral agency are not solely individual matters; instead, many societal factors affect nursing home staff members’ experiences of moral distress and whether or not they can act as moral agents.” |
RESULT
I am sorry, I don’t understand your result, because I miss categorisation. I think you have misunderstood my advice to look at Graneheim’s Figure 3. I meant the “column of subthemes” as a guidance how to abstract and formulate subthemes, to give meat to the bones of experiences. In the appendix you write: “Sub-themes (Further elaborated in text)” What I mean with categorisation is to abstract and elaborate the text into subthemes. In the appendix you repeat the following: Situations, circumstances and stakeholders are contributing to moral distress =sorting. You are still in the sorting phase. You need to reformulate them into subthemes according to Graneheim. After you have done this, I advice to reformulate the start of the result by listing your categorisation as a results. The following is just to illustrate what I mean with giving meat to the bones of the sorting (just borrowed content from Condensed meaning units in appendix):
Moral distress in nursing home care contained experiences of lack of time, high external demands and non-care related tasks hindering staff to give good care and handling aggressive residents and family members. This triggered staff to …..
Consider making Appendix to a true result table, for instance naming the subthemes Maincategories (as you write, that your categorisation is more at the manifest level) and listing sub-categories, as your text under each subtheme are very long.
Author Response
Thank you for your patience. Please see the attached document of our changes

Round 3
Reviewer 2 Report (New Reviewer)
Comments and Suggestions for Authors
Very good job with categorisation! The only strong advice I have now is to delete the two first columns in Table 2 (Meaning units Condensed meaning unit Description close to the text Condensed meaning units). These belong to method. And recommendation to only keep "categories" and "main categories" in the headings in the last two columns (that is, delete "Interpretation of the underlying meaning/"... (“What”) and (“How”).
Additionally just a minor comment is to change in the aim change "is" to "was" (you have finished the study).
This manuscript is a resubmission of an earlier submission. The following is a list of the peer review reports and author responses from that submission.
Round 1
Reviewer 1 Report
Comments and Suggestions for Authors
Thank you for the opportunity to review this manuscript, which addresses an important and understudied issue.
Comments for consideration:
Generally – The manuscript should be reviewed for grammar, it appears that English might be a second language for the authors. Such review might also be valuable in refining the results/presentation of results.
Abstract
- moral stress and moral distress are used interchangeably – do these reflect the same construct? In the introduction, distress is recognized as a subtype of stress. Perhaps use one term or the other in the abstract
Line 5 -“right to influence” I have not heard of this previously if related or subsumed under right to self-determination, perhaps remove
Introduction
Line 21-23 – definition of moral distress – does not attribute causality to organizational constraints. This is part of other definitions of moral distress – could be discussed? Later Jameton’s definition, which recognizes this is adopted (lines 89-91), perhaps it should be presented earlier.
It is unusual to present results (i.e. quotes) in the introduction
line 38-39 - - isn’t the behaviour grounded in the illness – i.e. the dementia?
Introduction/background – the criterion of capacity in the context of self-determination in this population is important and should be developed a bit further. It is likely that rather than purely self-determination, those with dementia can express preferences that should be honoured where possible and this is likely what is meant by “influence.” Can those with severe dementia truly express autonomy? Are there limits to free will in this population? For example, if the resident has significant skin breakdown, perhaps there is a limit to how long they can be left in a wet pad.
line 78-81 – it is not clear how those negative emotions can guide staff?
Lines 135-136 – and possibly employment repercussions/discipline?
Methods
Were the interviews conducted in English or Swedish? Who transcribed/translated? This should be stated
Line 180 -we excluded substitutes – what does this mean? Substitutes for what?
Line 184-185 – Why did staff indicate their study interest to the managers and not to the study staff directly? That is why was participation not kept confidential from management? Might staff’s contributions have been influenced by the manager knowing they were taking part in this study?
Results
– it might be helpful to further categorize Section 3.2 into subthemes, the section is long and hard to follow
-overall, quotes are more heavily drawn from groups 1 and 6, with little from groups 4 and 5 – was there a difference between the groups that warrants comment?
-noting this is a qualitative inquiry, were there any differing perspectives noted between the for-profit and municipally operated homes?
Line 262 – "residents should be able to express free will” isn't this this is contextual? they should always be allowed to express a preference, but free will might not be safe or practical – that is likely why they are dependent and in care
Section 3.2 – it would be helpful to know what the regulatory framework is for surrogate decision makers – suggest briefly addressing in the background section
Section 3.5 – it is not clear how this section/interpretation fits in a content analysis?
Line 426 and 480-491 can this moral agency can be labelled as passive? That is, the staff member acted actively and respected the resident wishes. That those wishes meant not to change the pad is the consequence of that action and moral agency. It is not “negligently following the residents wishes”
Second example – “telling staff right away” – is this an example of moral distress?
Discussion
line 438 – "affected" experiences of moral distress or caused these experiences?
line 439 – 441 poor grammar impacts meaning
Limitations – the limitations described with recruitment (manager identifying people) are at odds with the description of recruitment in the methods (research staff presented the study, interested staff told their managers)
Comments on the Quality of English Language
There is some awkwardness to the presentation that has likely arisen due to translation and English as a second language. It would be good to have this reviewed by someone with professional English proficiency. Indeed, it might be valuable to have someone with both professional English proficiency and qualitative expertise review the transcription/translation as well to ensure validity of the translation and interpretation
Author Response
Reviewer 1
|
Reviewer: 1 |
Changes |
Comments |
|
|
We reviewed the manuscript. |
We let an English-speaking person review the manuscript again. |
|
Abstract - moral stress and moral distress are used interchangeably – do these reflect the same construct? In the introduction, distress is recognized as a subtype of stress. Perhaps use one term or the other in the abstract
|
We changed the manuscript, and now there should be moral distress in the manuscript. |
We agree and decide to use just moral distress. |
|
Line 5 -“right to influence” I have not heard of this previously if related or subsumed under right to self-determination, perhaps remove |
We remove influence |
|
|
Introduction
Line 21-23 – definition of moral distress – does not attribute causality to organizational constraints. This is part of other definitions of moral distress – could be discussed? Later Jameton’s definition, which recognizes this is adopted (lines 89-91), perhaps it should be presented earlier.
|
We added one reference in the introduction that defined moral distress as psychological distress and experience of moral events. |
We agreed and improved the explanation of moral distress. |
|
|
We added some explanation as to why we started with quotes in the introduction. |
We understand that this way of using quotes is unusual, but we will try to have them in the introduction to illustrate the topics of the article. |
|
|
We added some words about behavior that also depend on the resident’s illness (dementia). |
We agreed and improved that relationship. |
|
Introduction/background – the criterion of capacity in the context of self-determination in this population is important and should be developed a bit further. It is likely that rather than purely self-determination, those with dementia can express preferences that should be honoured where possible and this is likely what is meant by “influence.” Can those with severe dementia truly express autonomy? Are there limits to free will in this population? For example, if the resident has significant skin breakdown, perhaps there is a limit to how long they can be left in a wet pad. |
We added text regarding persons with dementias’ possibilities to self-determination and related that to the Swedish context. |
We agree, and this is an important question. |
|
line 78-81 – it is not clear how those negative emotions can guide staff? |
We changed to affect. |
|
|
|
We added one paragraph that touches on employment. |
Nobody mentioned that in the interviews. |
|
Methods
Were the interviews conducted in English or Swedish? Who transcribed/translated? This should be stated
|
We added information regarding language. |
The first author conducted all interviews in Swedish, and the second translated the quotes into English. |
|
Line 180 -we excluded substitutes – what does this mean? Substitutes for what? |
We changed the world to temporary staff. |
|
|
Line 184-185 – Why did staff indicate their study interest to the managers and not to the study staff directly? That is why was participation not kept confidential from management? Might staff’s contributions have been influenced by the manager knowing they were taking part in this study? |
We clarified. |
Due to practical reasons, we have difficulties to meet all staff and those staff that was not attended the meeting was asked by the managers and the staff that conducted the meeting must have time to think about if they will participate and therefor they also tell the managers if they interested of participate. We mentioned in the limitation that managers could probably affect inclusions of staff |
|
Results
– it might be helpful to further categorize Section 3.2 into subthemes, the section is long and hard to follow
|
We divided the first section of the results into two categories. |
3.2 is a concise paragraph, however, we included more subheadings in section 3.1 |
|
-overall, quotes are more heavily drawn from groups 1 and 6, with little from groups 4 and 5 – was there a difference between the groups that warrants comment? |
|
Two groups stood out and had strong opinions on the subject (one from a private nursing home and one from a municipal nursing home). |
|
-noting this is a qualitative inquiry, were there any differing perspectives noted between the for-profit and municipally operated homes? |
|
No differences related to ownership. |
|
Line 262 – "residents should be able to express free will” isn't this this is contextual? they should always be allowed to express a preference, but free will might not be safe or practical – that is likely why they are dependent and in care |
We added one note about the regulations in Sweden. |
According to the Swedish National Board of Health and Welfare, nursing homes work for the resident’s autonomy and possibilities for self-determination. |
|
|
We added a paragraph about surrogate decision-making according to NBHW in the heading “ Nursing homes in Sweden.” |
We agree that this is important. |
|
|
We added text about how the figure was developed. |
The content analysis showed that there were factors at different levels, and the figure is an attempt to systematize and make these findings visible. |
|
Line 426 and 480-491 can this moral agency can be labelled as passive? That is, the staff member acted actively and respected the resident wishes. That those wishes meant not to change the pad is the consequence of that action and moral agency. It is not “negligently following the residents wishes”
|
We added a clarification. |
We agree that you can construct an agency differently. However, in this study, we will develop the notion further and frame the staff who did nothing actively to follow the residents' wishes as passive agents. The staff's duty is to let the residents decide according to NBHW 2012 but against the rules of the nursing homes. |
|
|
Yes, we consider, to inform fellow workmates is an active strategy. |
|
|
Discussion
line 438 – "affected" experiences of moral distress or caused these experiences?
|
We changed to caused. |
|
|
line 439 – 441 poor grammar impacts meaning |
We sent the manuscript to a language review again. |
|
|
|
We added some sentences regarding recruiting informants. |
We agreed and improved the text regarding recruiting informants. |
|
There is some awkwardness to the presentation that has likely arisen due to translation and English as a second language. It would be good to have this reviewed by someone with professional English proficiency. Indeed, it might be valuable to have someone with both professional English proficiency and qualitative expertise review the transcription/translation as well to ensure validity of the translation and interpretation |
We hand it over to the language reviewer. |
|
Reviewer 2 Report
Comments and Suggestions for Authors
Dear authors,
Your topic about elderly care and ethics is interesting and indeed a live issue in Sweden. However in this version of the manuscript I have a difficulty to understand the structure and message and above all, the resultsection needs categorisation! Particular guidance:
1. Introduction: It needs to be better structured and clear.
a) I wonder whether it might be a creative initiate to start with quotes, but I am uncertain where the quotes come from; from your own findings? This might be ok for a theoretical paper, but an empirical paper needs to be traditionally structured.
b) The part about moral distress is well formulated: “This study incorporates the concepts of moral distress and moral agency within a 68 theoretical framework. Lützén and Kvist (2012) differentiated among moral distress (psychological distress resulting from being unable to act according to one’s internal moral 70 guidelines), moral stress (resulting from moral demands and lack of control), and stress 71 of conscience (a philosophical perspective involving moral trouble).” However, you don’t seem to return to the distinction between moral distress and moral stress. You seem to use moral distress and moral stress haphazardly, for instance moral stress in Table 2 and in the abstract, otherwise moral distress. If there is an intention with this, please clarify.
2. Methods: This section needs structure, recommendation to use guiding subheadings.
a) You write you use content analysis according to Graneheim, but I lack description of your steps in analysis. The paper of Graneheim is not a method, just help to sort analysis terms. You seem to depart deductively from moral distress and moral agency. Please describe your analysis steps.
b) You need to clarify your definition of moral distress (the literature is full of different definitions).
3. Results: This section needs categorisation.
a) Your ambition in your aim is to explore experiences of moral distress, but you declare in the resultsection that you describe situations leading to moral distress.
b) Furthermore, what is gravely is the that your categories appear not to answer your aim about experiences neither situations. Instead, what I expect to be categories, your headings and subheadings appear to be some kind of sorting, not categorising.
c) The description of Figure 1 is “different levels”, but I lack explanations of what levels. To me it looks as you have used the socio ecological model. This needs to be clarified.
Comments on the Quality of English LanguageOk.
Author Response
|
Reviewer 2 |
|
|
|
1. Introduction: It needs to be better structured and clear. |
We added some explanation as to why we started with quotes in the introduction. |
We understand that this way of using quotes is unusual, but we will try to have them in the introduction to illustrate the topics of the article. |
|
b) The part about moral distress is well formulated: “This study incorporates the concepts of moral distress and moral agency within a 68 theoretical framework. Lützén and Kvist (2012) differentiated among moral distress (psychological distress resulting from being unable to act according to one’s internal moral 70 guidelines), moral stress (resulting from moral demands and lack of control), and stress 71 of conscience (a philosophical perspective involving moral trouble).” However, you don’t seem to return to the distinction between moral distress and moral stress. You seem to use moral distress and moral stress haphazardly, for instance moral stress in Table 2 and in the abstract, otherwise moral distress. If there is an intention with this, please clarify. |
We decided to use moral distress only. |
We agree that it could have been clearer. |
|
2. Methods: This section needs structure, recommendation to use guiding subheadings. |
We have written about deductive content analysis in more detail. |
We agreed and have improved the description and included more details. |
|
b) You need to clarify your definition of moral distress (the literature is full of different definitions). |
We clarify our definition of moral distress as both psychological and moral events according to Morley et al. 2019. |
We decided to use moral distress mainly according to Morley et al. 2019. |
|
Results: This section needs categorisation. |
We change the aim for the study to: The overall aim of the study is to explore how staff experience situations and factors leading to moral distress in nursing homes when working to promote residents’ self-determination |
The overall aim is changed to fit the study better. Moreover, the questions are answered in the results section “The core focus is to understand a) circumstances that may lead to moral distress and b) how moral distress can affect the moral agency of staff when they work to pro-mote residents’ self-determination“
|
|
b) Furthermore, what is gravely is the that your categories appear not to answer your aim about experiences neither situations. Instead, what I expect to be categories, your headings and subheadings appear to be some kind of sorting, not categorising. |
We have further developed the paragraph about the analysis according to Graneheim and Lundman (2003) and adjusted the aim as you see above. |
|
|
c) The description of Figure 1 is “different levels”, but I lack explanations of what levels. To me it looks as you have used the socio ecological model. This needs to be clarified. |
We consider it as a result; we clarify this by the model. However, we added a paragraph about the analysis leading to this model. |
We worked inductively with this, and now, in retrospect, it is reminiscent of the social-ecological model. |
Reviewer 3 Report
Comments and Suggestions for Authors
Moral distress is assumed to be an individual concept and it seems that there are no social components that are linked to this phenomenon. Jameson himself refers, in his definition, to what one knows to be right. This fact is not, in any case, individual so that one could incur in fallacy of affirmation of the consequent. Let me explain. As the individual suffers the distress, we assume that it is individual. I repeat this is a fallacy. For this reason, I suggest that the authors make mention in the introduction of these aspects, although, later, they study it from a psychological framework.
On the other hand, in section 1.1. mention is made of some previous studies. When approached from this perspective, it gives the impression that moral studies have been developed "de novo". Therefore, I suggest a brief review of the main moral (not ethical) theories and a mention of the theory to be followed in the subsequent analysis.
The methodology is well developed, as are the results. In this sense, I consider the work to be excellent.
The conclusions are timely.
The bibliography is good, although it presents theoretical deficiencies as I mentioned before.
Author Response
|
Reviewer 3 |
|
|
|
Moral distress is assumed to be an individual concept and it seems that there are no social components that are linked to this phenomenon. Jameson himself refers, in his definition, to what one knows to be right. This fact is not, in any case, individual so that one could incur in fallacy of affirmation of the consequent. Let me explain. As the individual suffers the distress, we assume that it is individual. I repeat this is a fallacy. For this reason, I suggest that the authors make mention in the introduction of these aspects, although, later, they study it from a psychological framework. |
We added one reference that further defined moral distress as both psychological distress and experience of moral events and clarified that we adopted this perspective study. |
We agreed it is essential. |
|
On the other hand, in section 1.1. mention is made of some previous studies. When approached from this perspective, it gives the impression that moral studies have been developed "de novo". Therefore, I suggest a brief review of the main moral (not ethical) theories and a mention of the theory to be followed in the subsequent analysis. |
We added one reference to general moral theory and one systematic review that further defined moral distress. |
There is no space to write a review of moral theory in the article. However, we have added a sentence that touches on general moral theory. |
|
The methodology is well developed, as are the results. In this sense, I consider the work to be excellent. |
|
|
|
The conclusions are timely. |
|
|
|
The bibliography is good, although it presents theoretical deficiencies as I mentioned before. |
|
|
Round 2
Reviewer 2 Report
Comments and Suggestions for Authors
I am sorry, but you had a chance to make this manuscript coherent with adequate theory and analysis of data. I still have concern about:
1. A synthesis of multiple definitions of moral distress and what definition you depart from in your named analysis is lacking.
2. I am afraid that using quotes from your interview data to illustrate moral distress in the Introduction does not make sense. My judgment is that this quote does not cover the phenomena of moral distress: "When you’ve worked for this long, you are not as affected by it. It flows off you, that 30 they call you ugly words, can suddenly spit on you. So they become aggressive, they 31 can tear my skin and all this. It’s hard at the beginning. And it almost always has to 32 do with stress. (No. 1)" I interpret this to reflect stress in own work situation, not a moral distress in concern of the vulnerable with less power than the staff. I think this is serious.
3. The method section is still very unstructered and difficult to follow. You have added the word 'deductive' in the analysis section: "The interviews were interpreted and analyzed via a deductive content analysis in line with the theory of moral distress and moral agency." First, there is no THE theory and I lack reference to which definition you have used. Second, it is not clear which analysis method you have used. Graneheim and Lundman do not describe using deductive analysis. Your adding of "In line with Graneheim and Lundman (2003), we identified meaning units, condensed them, and assigned them codes. We sorted the different codes into categories and then into the two themes of moral distress and moral agency." is not description of deductive analysis. Inductive analysis is described (you may look at their 2017 paper, mentioning abduction).
4. The results/categorisation do still not respond to your new aim. The themes and categories should answer the aim how staff experience situations and factors leading to moral distress in connection to self-determination. First, I cannot find the two themes that should be described in the beginning of the results. Second, I am not able to identify abstraction of experiences in the form of categories. To me your headings still look as just sorting. T
Reviewer 3 Report
Comments and Suggestions for Authors
Some previous conceptual problems have been clarified. In this regard, I am grateful for the effort made by the researchers. However, some issues still need to be modified.
Dealing with a subject as complex as ethics and morality implies the need to show prior knowledge of the numerous works carried out over the centuries. Obviously, the work presented here refers to the current context, but this does not imply the absence of any mention of the entire history of ethics. This aspect is undoubtedly the greatest weakness of this study. In fact, it does not contemplate the entire research effort of the human sciences in this particular field. Undoubtedly, this must be remedied.
This has improved the article but the fallacies in the paper have not been adjusted. It is essential to include social aspects related to ethics and morality. Moral distress (by definition) is social, therefore it is necessary to talk about the social aspects that affect people. Likewise, there is no mention of the different ethical currents (e.g. utilitarianism, personalism, socialism, idealistic ethics, virtue ethics, etc.). It is necessary to show that these perspectives are known and to take sides with one of them. Otherwise, it seems as if ethics and morality had not been studied since classical Greece.
Therefore I cannot recommend acceptance of the article as it has not addressed all previous suggestions.